# Do "English" Named Entity Recognizers work well on Global Englishes?

**Alexander Shan**, **John Bauer**, **Riley Carlson** and **Christopher D. Manning**
Department of Computer Science
Stanford University
Stanford, CA 94305-9030, U.S.A.
{azshan, horatio, rileydc, manning}@stanford.edu

## Abstract

The vast majority of the popular English named entity recognition (NER) datasets contain American or British English data, despite the existence of many global varieties of English. As such, it is unclear whether they generalize for analyzing use of English globally. To test this, we build a newswire dataset, the *Worldwide* English NER Dataset, to analyze NER model performance on "low-resource" English variants from around the world. We test widely used NER toolkits and transformer models, including RoBERTa and ELECTRA, on three datasets: a commonly used British English newswire dataset, CoNLL 2003, a more American-focused dataset, OntoNotes, and our global dataset. All models trained on the CoNLL or OntoNotes datasets experienced significant performance drops—over 10% F1 in some cases—when tested on the Worldwide English dataset. Upon examination of region-specific errors, we observe the greatest performance drops for Oceania and Africa, while Asia and the Middle East had comparatively strong performance. Lastly, we find that a combined model trained on the Worldwide dataset and either CoNLL or OntoNotes lost only 1–2% F1 on both test sets.

## 1 Introduction

Most of English Named Entity Recognition (NER) uses American or British English data, with less attention paid to low-resource English contexts. Multiple problems may occur in low-resource NER settings; for example, named entities with region-specific meanings can be confused for common words. Indeed, the Japanese Diet is a governmental body, but NER models focused on US and British English may incorrectly interpret this entity as a medical term.

Among many NER datasets released in recent years,[1] the most widely used datasets are CoNLL

2003 (Tjong Kim Sang and De Meulder, 2003) and OntoNotes (Weischedel et al., 2013), which focus on British and American English, with significant European Parliament coverage. Other recently created NER datasets study the medical domain, such as the n2c2 challenges (Henry et al., 2019), historical English (Ehrmann et al., 2022), or music recommendation terminology (Epure and Hennequin, 2023), still using American and British English. The lack of regional variety in these datasets suggests that models trained on these datasets might not accurately recognize entities from more global contexts. Furthermore, the lack of test data for other regions makes it difficult to even measure this phenomenon.

In this work, we evaluate the performance of a variety of NER tools, including Flair and SpaCy on this dataset. We then retrain two commonly used NER models, those of Stanza (Qi et al., 2020) and CoreNLP (Manning et al., 2014), to see whether their performance improves when the Worldwide English dataset is combined with their usual training data. Using these toolkits allows for easy retraining while comparing the performance losses between models using categorical features, word embeddings, and transformers. When using Stanza, in order to compare the generalization of different language models, we retrain with each of Roberta-Large (Liu et al., 2019), Electra-Large (Clark et al., 2020), and just the CoNLL 2017 word vectors (Ginter et al., 2017).

We find that, as expected, there is a large dropoff in performance when the models are applied to regions not well covered by the training data in CoNLL or OntoNotes. This is true whether the model uses categorical features, word vectors, or transformer language models. When considering specific regions around the world, we find that Indigenous Oceania and Africa had the worst performance due to a disproportionately large number of unrecognized tokens. We believe these

---

[1] A collection of NER references is available at https://github.com/juand-r/entity-recognition-datasets

results reflect the impact of minimal discussions of the Global South in U.S. and U.K. newswire. Across all regions, models trained on CoNLL or OntoNotes struggled to accurately identify cultural and currency names. Furthermore, we find that retraining the models in question on the training split of the new dataset improves their performance on those regions, but has the opposite effect of hurting the performance on US and British English texts. When retrained on both an existing dataset and the new dataset at the same time, however, the models maintain their previous performance on CoNLL or OntoNotes while drastically improving their performance on other worldwide regions. We make our new dataset public to better represent the diversity in written English around the world.[2]

## 2 Related Work

NER has been explored in non-English languages with great success, with one example being Stanford NLP's open-source Stanza NER model (Qi et al., 2020; Zhang et al., 2021), which covers many languages such as Chinese and Arabic, along with including several other English medical models. There has also been research on low-resource foreign languages with notable success using neural-based approaches (Cotterell and Duh, 2017; Adelani et al., 2022). However, low-resource contexts of English have not been recently examined. Louis et al. (2006) showed that Western-English trained models performed poorly in South African contexts of English due to the presence of unknown words such as "Xabanisa", or alternative uses of words like "Peace", a common South African name. However, their model was a simple Bayesian network, so their results are not indicative of how modern neural models might perform, especially with pretraining. Ghaddar et al. (2021) revealed that name-regularity bias still exists in neural models, even when contextual clues are present. For example, in the sentence "Obama is located in southwestern Fukui Prefecture.", state-of-the-art NER models labeled "Obama" as a person rather than a location. Ghaddar et al. did not examine contexts of English, so further research remains important.

Outside the NER realm, Ziems et al. (2023) explored dialectic differences in machine translation and question answering, but did not address NER.

[2]The dataset is available at https://github.com/stanfordnlp/en-worldwide-newswire

## 3 Dataset Sourcing and Construction

### 3.1 Sourcing

The Worldwide NER dataset is composed of 1075 articles containing 674,000 tokens sourced from countries and regions across the globe where English is used. Articles were taken from a variety of press organizations per country, with organizational diversity scaling with the number of texts from a given country. For subsequent testing reasons, we separate texts into regional buckets, partitioning the corpus into articles from Asia, Africa, Latin America, the Middle East, and Indigenous Commonwealth (indigenous Oceania and Canada). Appendix B contains a breakdown of the regions and their number of articles.

### 3.2 Labeling

We partnered with data labeling platform Datasaur as well as an annotation workforce provided by third-party annotation service MLTwist. MLTwist contracted a Ghana-based labeling service, Aya Data, for the annotations. The annotators were fluent English speakers and each labeling batch was annotated by one labeler before review from a member of MLTwist and our research team. When reviewing, MLTwist and the research team would sync to have a consensus on each annotation.

To assess our labeling quality, we use Cohen's Kappa scores as reported by the Datasaur platform. The combined score when comparing all of the individual annotators with the review process described above is 77.47. While scales for Cohen's Kappa are heavily context-dependent, we believe this indicates a thorough labeling process.

Our Worldwide English dataset uses 9 classes: Date, Person, Location, Facility, Organization, Miscellaneous, Money, NORP (national, organizational, religious, or political identity), and Product. Appendix A contains a full list of our dataset's labels, accompanied with definitions. We condensed the 18-class label framework of OntoNotes 5.0 into 9 classes because we wanted to isolate the classes that would pose challenges across English contexts. Since classes like *Percent*, *Ordinal*, and *Time* are statically used across English contexts, we omitted most numeric and time expressions. Some classes appeared too infrequently to justify a new class, like *Work of Art* and *Law* which were condensed into the MISC class, similar to CoNLL03.

Additionally, the data labeling instructions were designed to align with that of the OntoNotes 5.0

| Embedding | Train | CoNLL | All WW | Africa | Asia | Indigenous | Latin America | Middle East |
|---|---|---|---|---|---|---|---|---|
| w2v + char | CoNLL | 91.02 | 77.35 | 77.19 | 79.68 | 69.80 | 77.95 | 75.20 |
| Electra | CoNLL | 93.18 | 82.94 | 83.35 | 84.92 | 76.90 | 82.51 | 81.10 |
| Roberta | CoNLL | 92.54 | 83.29 | 83.11 | 85.19 | 76.12 | 83.24 | 82.75 |
| w2v + char | Worldwide | 70.67 | 87.19 | 86.39 | 89.71 | 82.60 | 87.17 | 85.28 |
| Electra | Worldwide | 75.06 | 90.11 | 89.66 | 92.20 | 84.08 | 89.93 | 89.27 |
| Roberta | Worldwide | 75.69 | 90.23 | 89.59 | 92.18 | 86.83 | 90.16 | 88.79 |
| w2v + char | Combined | 90.72 | 86.48 | 85.59 | 89.15 | 81.15 | 86.45 | 84.70 |
| Electra | Combined | 92.82 | 89.99 | 89.32 | 92.39 | 84.07 | 89.55 | 89.13 |
| Roberta | Combined | 91.99 | 90.10 | 89.27 | 92.30 | 87.28 | 89.66 | 88.67 |

Table 1: Stanza entity F1 Scores on CoNLL and Worldwide test data when trained on CoNLL, Worldwide, or Combined training data. Combined training gives close to best results in all cases.

| Entity | CoNLL | All WW | Africa | Asia | Indigenous | Latin America | Middle East |
|---|---|---|---|---|---|---|---|
| LOC | 72.96 | 91.44 | 90.91 | 92.49 | 86.69 | 91.53 | 91.22 |
| MISC | 65.16 | 84.81 | 80.92 | 88.75 | 83.43 | 84.70 | 84.26 |
| ORG | 63.07 | 86.55 | 86.78 | 90.02 | 80.00 | 84.82 | 83.79 |
| PER | 94.88 | 96.02 | 96.08 | 96.65 | 95.60 | 96.30 | 94.45 |
| Overall | 75.69 | 90.23 | 89.59 | 92.18 | 86.83 | 90.16 | 88.79 |

Table 2: Stanza entity F1 scores when trained using Roberta on Worldwide

dataset for consistency. The only exception was Date, where we did not label generic time expressions such as "the last three months", which receive a label in OntoNotes. Unlike OntoNotes, we use nested NER annotations when relevant, so that a phrase such as "2022 FIFA World Cup" has the entire text labeled MISC to represent the event, "2022" labeled Date, and "FIFA" labeled Organization.

### 3.3 Preprocessing

To preprocess the data, we convert the annotations into BIOES format. BIOES marks tokens with their class and position in a named entity span as one of beginning, intermediate, ending, or singular, with non-named-entity words labeled O.

When comparing with CoNLL-based models, such as CoreNLP, Flair, and Stanza's 4 class model, to maintain consistency when building combined models and comparing results, we collapsed the annotations for the Worldwide English data with 9 classes into 4 classes (person name, location, organization, and MISC). A complete explanation of the collapsing process is outlined in Appendix D. For CoNLL03, we use the original version and its defined train, dev, and test sets. For OntoNotes, we use the train/dev/test split from Weischedel et al.

(2013). For our own dataset, we perform a 70/10/20 train/dev/test split, randomized within each region using stratified sampling.

## 4 Experiments using Stanza

We first examine Stanza models (Qi et al., 2020) on CoNLL and our Worldwide English; we consider other NER systems and OntoNotes in section 7.

### 4.1 Approach

We train and test Stanza's NER model with three different pre-training configurations (word2vec word embeddings and character model only, RoBERTa embeddings, ELECTRA embeddings). We train models on CoNLL03 and the Worldwide English dataset separately. We also consider a combined model trained on both CoNLL03 and our dataset. During testing, we compare the models for performance on both the CoNLL03 and Worldwide datasets, evaluating on the joint test set of all regions along with region-specific results.

**Availability** The OntoNotes model used in the following experiments is not ideal for public release, as it loses the granularity of the 18 classes provided by OntoNotes. Instead, we used the 9

class Worldwide data to finetune the hidden representations of the 18 class OntoNotes model; the model is available through Stanza.

The source code for converting the raw dataset to regional and combined portions, training a Stanza model using multiple data sources, converting OntoNotes to 9 classes and our data to 4 classes, and reproducing the final combined model are available in the latest Stanza release.[3]

**Hyperparameters** We train using stochastic gradient descent (maximum gradient descent steps = 200,000, with early termination) and then score the model against the dev and test sets of CoNLL03 and the Worldwide dataset. We use a learning rate scheduler to decrease the learning rate after a plateau, with the early termination condition of the learning rate reaching 0.0001. Since our goal was to examine the differences between models trained on the various datasets, and not to maximize performance, we use the default hyperparameters provided by Stanza. Training took around 3 hours for each dataset on an Nvidia 3090, with the exception of the combined model (5 hours). Values for hyperparameters are shown in Appendix C

### 4.2 Overall Results

The results in Table 1 for the Stanza model tested on CoNLL03 are consistent with other state-of-the-art models tested on CoNLL03 (Lim, 2023). As expected, the model's performance suffered when tested on the opposite dataset from which it was trained. While this effect is expected, the degree to which the model confuses tags is significant considering that it was supported with RoBERTa and word2vec, which included worldwide training sources like Wikipedia. The word2vec-based model with transformer performs significantly worse, 5 F1, than its transformer counterparts, suggesting that the improved understanding of words from context and a larger pre-training set makes a difference. Additionally, the combined model performs well across both datasets; it is competitive with the solely Worldwide English-trained model on the Worldwide English evaluation sets and outperformed the CoNLL03-trained model on its respective evaluation sets. As for specific language regions, the CoNLL03-trained model performed worst in indigenous language contexts, followed by African contexts. The other regions (Asia, Latin America, Middle East) respectively shared similar

performance to each other. As shown in Table 2, we find that the model trained on Worldwide performs well on PER, including on the original CoNLL03 dataset. The other categories, especially ORG and MISC, show larger degradation. Below, we discuss specific error cases from each region.

## 5 Analysis

To better understand model behavior in error cases, we examine the sentences with incorrect predictions, generalizing error patterns across all regions and examining region-specific challenges.

### 5.1 Universal error patterns

A large challenge for the model(s) trained on CoNLL03 and tested on our Worldwide dataset was currencies. CoNLL03 labels currencies as MISC, such as "A$", "US$", and "C$" for currencies; we find hundreds of instances where other currencies were incorrectly labeled "O" by the CoNLL03-trained model. Upon inspection of CoNLL03, we diagnose the error as the result of currencies such as the yen, zlotys, and lei incorrectly labeled "O" in training. Consequently, currencies in our dataset such as the rupee, baht, colones, and riyal were often missed as MISC. Another error in many regions was the improper labeling of names. Take the following example for instance:

*President* **Yoon Suk - yeol** *ordered on Monday the preparation of a roadmap...*

*Yoon Suk - yeol* is a full name, yet the model incorrectly labeled *Yook* and *yeol* as S-PER, leaving *Suk* as a O tag instead of I-PER. This error is a consequence of the differences in name structure across the world, as is discussed in section 5.2.

### 5.2 Regional errors

Next, we analyze error patterns that frequently appeared in specific regions. Every region had some share of these errors, but we highlight the regions where a type of error was overwhelmingly present.

**Asia** As described in section 5.1, Asia had the majority of person name errors, whether it be I-PER tags incorrectly labeled O, or E-PER tokens labeled O. This error is strongly correlated with the structure of names in parts of Asia rather than the tokens being unrecognizable; names in CoNLL03 frequently take on the structure of {first, last} and not {first, middle, last}. However, in other English contexts, names with hyphens to join 3 or more

---

[3]<https://stanfordnlp.github.io/stanza/>

morphemes can be commonplace (e.g. *Chen Mei - hsiu*). Hence, for the model, these names appear as multiple single names listed in sequential order, rather than a single named entity.

**Africa**  African texts generally had below-average performance amongst the Worldwide corpus, but had disproportionately high error count when the gold tag was MISC and the predicted label was ORG. These errors were found in sentences where some tokens were unrecognized and lacked strong context clues. Consider the following example, for instance:

*Directed and produced by Kwabena Gyansah ( **Azali** ) , **High Currency** also stars Brihanna Kinte ( **Ghana Jollof** ) , Edward Kufuor ( **Accra Medic** ) , Kweku Elliot ( **To Have and To Hold** ) , ...*

Each bolded entity is a Work of Art (properly MISC), but was labeled as ORG. While *High Currency* contextually seems like a film production, the remaining named entities could be mistaken for groups. Some of the tokens are missing from training and pre-training, leaving the model to classify unknown words amid unclear contextual clues, leading to these errors. The models also exhibit errors amid clear contexts, like in this case:

*The a la carte restaurant run by executive chef Ulric Denis is pricy, in a delightful, luxurious but somewhat impersonal surrounding and can offer some of the island's best examples of modern cooking with a true **Seychellois** taste .*

*Seychellois*, a NORP (and therefore MISC in CoNLL parlance) describing the Seychelle people, is mistaken for an ORG. Despite the strong contextual clues hinting that the token is a MISC, the model is confused because *Seychellois* was an unknown word, missing during training. Hence, we observe that the contextual signal to the model was overpowered by the confusion induced by the presence of this unknown token. As a general pattern, we find that when sentence context is uninformative, unknown tokens are challenging for models, yet this is not solved with clear context either.

**Indigenous**  The indigenous region, composed of Oceanic and Canadian native tribal news, posed the greatest challenge to the CoNLL03-trained model. We find that the large share of unknown tokens from these texts induced great confusion for the models. Similar to the African corpus, we observe unrecognizable tokens negatively impacting model performance regardless of contextual clues.

**Example:**  *With Prime Minister Anthony Albanese's attendance at **Garma**, a great sense of elation that maybe, perhaps, something might be about to change has taken hold.*

Garma, a cultural festival for indigenous North Australians (MISC) was incorrectly labeled LOC. Contextual clues signal that Garma is a location, given that Albanese attended "at" Garma. Furthermore, it typically shows up in the Wikipedia data used by Roberta and ELECTRA as a location. However, with a stronger understanding of the word, it would be clear that Garma is referred to as an event (MISC) in this sentence. Indeed, if Garma is replaced with Thanksgiving, an American tradition, the model properly identifies the named entity. This example shows how unknown tokens cause models to decide labels based on potentially misleading contextual reasoning. The remedy to this issue is having stronger signals from word meanings, which is achievable through greater inclusion of broadly sourced training data.

**Latin America**  Our models had an average performance on the Latin America corpus compared to the other regions, but we observe some interesting error patterns. We once again see errors associated with unfamiliar tokens, such as the following:

**Example:**  *Simón \* is a FARC ex - combatant living in the Icononzo camp ( ETCR Antonio Nariño ) in the **Andean** region of Tolima . " I don ' t want to live in fear for another four years , " he said .*

Despite Andean (LOC) referring to the Andes Mountain Range, it was mistakenly labeled MISC, likely because it was misinterpreted as a NORP. That is, if one replaces Andean with a NORP, such as American, the sentence makes sense. Hence, since Andean was an unfamiliar word, the model turned to unreliable contextual clues for guidance, resulting in error. Next, we observe a case of name-recognition bias in a second example:

**Example:**  ***Brazilian Amazon** deforestation doubled from the 2009-2018 average, with 22 percent more forest lost in 2021 than the previous year.*

While the Brazilian Amazon is clearly a location, the model incorrectly predicted MISC. We find that the model has a bias for *Brazilian* as a NORP (MISC), inducing the erroneous prediction. This is a unique error, as *Brazilian* appears in training and pre-training, but not with the same meaning as it has in this sentence. Hence, we observe that the lack of contextual diversity in training caused the sentence context signal to be overwhelmed by the

| Train | Test | All | LOC | MISC | ORG | PER |
|-------|------|-----|-----|------|-----|-----|
| CoNLL | CoNLL | 89.27 | 91.41 | 81.60 | 85.63 | 94.07 |
| CoNLL | Worldwide | 70.47 | 79.65 | 59.87 | 58.75 | 76.25 |
| Worldwide | CoNLL | 63.67 | 71.10 | 50.03 | 38.28 | 83.28 |
| Worldwide | Worldwide | 82.65 | 84.92 | 76.12 | 75.78 | 90.23 |
| Combined | CoNLL | 88.86 | 91.10 | 77.75 | 85.36 | 94.83 |
| Combined | Worldwide | 83.04 | 85.95 | 76.36 | 75.45 | 90.63 |

Table 3: F1 Scores for CoreNLP by type

| Train | All | Africa | Asia | Indigenous | Latin America | Middle East |
|-------|-----|--------|------|------------|---------------|-------------|
| CoNLL | 70.47 | 72.13 | 72.67 | 60.87 | 71.47 | 66.88 |
| Worldwide | 82.65 | 82.12 | 85.06 | 76.12 | 81.04 | 81.23 |
| Combined | 83.04 | 82.12 | 85.73 | 76.56 | 81.58 | 81.73 |

Table 4: F1 Scores for CoreNLP on Worldwide by region

model's learned meaning for *Brazilian*.

**Middle East**  The Middle Eastern corpus had comparatively strong performance among the Worldwide texts. Of course, there were several notable error patterns, such as improperly labeled person names and currency names. Additionally, we find many error cases induced by unknown tokens.

**Example:**  *Making the remarks in an interview with the state-owned broadcaster, **TRT Haber**, ...* TRT Haber is a Turkish news agency (ORG). However, it does not show up as an entity in either CoNLL or OntoNotes. Considering the CRF model, the word *interview* should be a strong hint to the model. It occurs in the original datasets in contexts near both PER and ORG, but in this case, the model incorrectly labels TRT Haber as PER.

## 6  Alternate Explanations

Although our hypothesis is that regional variations are the largest contributor to performance gaps between models, several other possibilities could be valid. We address some possible explanations here.

**Dataset Domain**  One possible explanation of the scores would be different domains. However, CoNLL is trained entirely on newswire, and the majority of OntoNotes is also newswire. Therefore, aside from regional or temporal differences, the domains of the datasets should be very similar. Differences in annotation style could also be a culprit. However, except where noted for the Date

label and the collapsing of Location and GPE, we used CoNLL and OntoNotes as guidelines for the annotation team to follow.

**Temporal Drift**  A concerning problem would be temporal drift in annotations. We use a dataset similar to CoNLL to demonstrate that temporal drift alone does not account for the differences seen (Liu and Ritter, 2023). This dataset uses CoNLL style annotations, but with news articles from 2020. Liu and Ritter found that pretrained embeddings such as Roberta do not degrade in performance when trained on CoNLL and tested on 2020 data.

Here, we instead use the 2020 data as additional training data, using Stanza with the Roberta embedding. We find that there is some improvement in the Worldwide test score from adding the 2020 data, but the regionally appropriate data from Worldwide has substantially more impact.

| Training | CoNLL | WW |
|----------|-------|-----|
| CoNLL | 92.54 | 83.29 |
| CoNLL + WW | 91.99 | 90.10 |
| CoNLL + 2020 | 92.52 | 84.57 |
| CoNLL + 2020 + WW | 91.95 | 89.75 |

**Additional Data**  As we saw, adding some amount of 2020 data improved the scores on the Worldwide test set, even when not regionally targeted. We perform an ablation study to show that having regionally appropriate training data is more relevant than simply adding more training data.

In particular, we choose one of the regions, Asia, and train Stanza with Roberta embeddings on increasingly large subsets of CoNLL and Worldwide

| Model | CoNLL | Worldwide | Africa | Asia | Indigenous | Latin America | Middle East |
|---|---|---|---|---|---|---|---|
| ner-fast | 91.45 | 78.70 | 79.70 | 80.27 | 72.43 | 79.37 | 75.43 |
| ner | 91.59 | 79.36 | 80.29 | 80.94 | 72.80 | 80.00 | 76.37 |
| ner-large | 91.71 | 85.81 | 86.40 | 86.93 | 80.79 | 85.31 | 85.13 |

Table 5: Entity F1 Scores for Flair by region

| Test \ Train | OntoNotes | | | Worldwide | | | Combined | | |
|---|---|---|---|---|---|---|---|---|---|
| | w2vec | Roberta | Electra | w2vec | Roberta | Electra | w2vec | Roberta | Electra |
| OntoNotes | 89.88 | 91.43 | 92.55 | 74.35 | 77.26 | 77.86 | 89.26 | 90.80 | 91.62 |
| Worldwide | 71.62 | 76.92 | 77.00 | 86.04 | 89.31 | 88.97 | 83.62 | 86.86 | 87.79 |

Table 6: Entity F1 Scores for Stanza on OntoNotes

excluding the Asia portion of the training data. We find that although the Asia dataset is a fraction of the entire dataset, training on just CoNLL and Asia is more accurate than the larger subsets. These results are summarized in table 7.

**Data Leakage** Another explanation would be that of data leakage from a regional training sets to the test set, especially in the case of a news story with worldwide coverage. For example, if an article from Latin America covered an event concerning an American presidential election in the training data, an article from a news source in Asia covering the same event might be randomly assigned to the test set, giving an unfair advantage to a model trained on the Worldwide dataset. The data collection emphasized stories of regional interest, however, and so there should not be data leakage of international news stories.

**Alternate Embeddings** An embedding with more worldwide coverage might also show less degradation. For example, Multilingual Bert (Devlin et al., 2018) and XLM-RoBERTa-Large (Conneau et al., 2019) both include multilingual text, perhaps including many of the terms within our Worldwide dataset. Instead, we find that they perform no better than Roberta-Large or Electra-Large. Meanwhile, DeBERTaV3 (He et al., 2023) is more accurate, but still exhibits degradation.

| Emb | Training | CoNLL | WW |
|---|---|---|---|
| DeBERTaV3 | CoNLL | 93.03 | 83.64 |
| XLM-RoBERTa | CoNLL | 92.54 | 82.53 |
| mbert | CoNLL | 92.09 | 81.44 |
| DeBERTaV3 | WW | 75.17 | 90.20 |
| XLM-RoBERTa | WW | 75.46 | 90.25 |
| mbert | WW | 75.32 | 89.20 |

## 7 Experiments using Other Models

### 7.1 CoreNLP on CoNLL

We used Stanford's CoreNLP package (Manning et al., 2014), an NLP software distribution written in Java before neural models became widespread, to test the effectiveness of its model on the worldwide NER data. Our results are shown in table 3 for class-based scores and in table 4 for scores by region.

CoreNLP includes a model trained on CoNLL, with the caveat that it does not use B- and I- tags, but rather assigns labels as either part of a class or O for outside. This is rarely a practical issue in English, with exceptions such as lists without commas. The model is a CRF which uses text features of the words and their neighbors. For example, the word "CoreNLP" would itself be a feature, along with the prefixes "C", "Co", ..., and the suffixes "P", "LP", .... Similarly, "CoreNLP" and other categorical features of the word would be used for features for the neighboring words. The baseline CoNLL model included with CoreNLP has an F1 of 89.27% on CoNLL03. (Finkel et al., 2005)

Testing the included model on the Worldwide dataset shows a very large drop in performance, 70.47% F1. The largest drop was on Organization, which went from 85.63% to 58.75% F1.

We retrained the model on the Worldwide dataset, achieving an F1 of 82.65%. The Worldwide trained model also suffered a very large drop when tested on the CoNLL dataset, falling to 63.67% F1. Again, Organization was the largest drop, going from 75.78% to 38.28% F1.

A likely explanation for the large drop in performance is that categorical features can provide con-

| Training portions | Training tokens | Entity coverage | Asia Worldwide F1 |
|---|---|---|---|
| CoNLL | 203621 | 31.65 | 85.19 |
| CoNLL + Latam | 307537 | 35.49 | 87.75 |
| CoNLL + Latam + Africa | 409432 | 38.02 | 88.18 |
| CoNLL + Latam + Africa + ME | 468788 | 40.97 | 88.06 |
| CoNLL + Latam + Africa + ME + Ind. | 521580 | 41.04 | 88.51 |
| CoNLL + only Asia | 351792 | 55.16 | 90.36 |
| CoNLL + entire Worldwide | 669751 | 56.90 | 92.30 |

Table 7: Test scores for Asia in Worldwide NER on progressively larger subsets of the data. (Ind. = Indigenous)

text for common English words, such as in a sentence "(PERSON) works at (ORGANIZATION)", but the named entities themselves are completely different. For example, a local organization "Nuluujaat Land Guardians" starting with "Nuluujaat" does not resemble any organizations in CoNLL, and so the CoNLL trained model fails in the sentence "A recent legal move by the Naluujaat Land Guardians . . . to sue Baffinland Iron Mines . . . ". Instead, the capital letters are enough of a feature that the CoNLL trained model identifies it as a MISC.

Person and Location, however, are much more robust when tested across datasets. For Person, especially, even when the last names are different, many regions have similar first names. For example, Anne, Anthony, and Tom are common first names in the Indigenous test set.

Finally, we retrained the model on a combination of the CoNLL and Worldwide datasets. This model mostly combined the strengths of the two previous models, achieving 0.8304 F1 on the Worldwide dataset and 0.8886 on CoNLL. The number of features in the trained models increased from 658K for the CoNLL model and 1M for the Worldwide model to 1.4M for the combined model; this indicates there is very little overlap between the features for the two models, resulting in the model effectively learning the two datasets in parallel.

When compared on a regional basis, we find that each model shows the same drop on Indigenous regions, with the original CoreNLP CoNLL model especially inaccurate on the Worldwide Indigenous section. Considering there is some Middle Eastern text in the original CoNLL, the CoreNLP model also does surprisingly poorly on the Middle East region.

## 7.2 Flair on CoNLL

Another widely used software package for NLP, especially for NER, is Flair (Akbik et al., 2019). The base English NER model for Flair makes extensive use of character embeddings (Akbik et al., 2019), whereas the higher accuracy model uses a transformer. Their models also use document-level features (Schweter and Akbik, 2020).

We performed experiments on the publicly available Flair models, reported in table 5. We note that as there are different versions of CoNLL in existence, it is not surprising that our results for Flair on CoNLL are not the same as reported, such as 91.71 instead of 94.09 for ner-large.

As expected, the transformer model performs much better than the basic models. In fact, the transformer model performs significantly better than either Stanza with a transformer or CoreNLP. We still note a large drop in performance on the Indigenous text, further confirming that this is the most difficult region to process. Furthermore, the results of ner-large on Worldwide are not as accurate as Stanza when trained specifically on the Worldwide dataset, suggesting that there would be gains in accuracy on the Worldwide dataset from retraining the Flair transformer model on a combined dataset.

## 7.3 Stanza on OntoNotes

We also trained the Stanza model for OntoNotes, see Table 6. Whereas for CoNLL, we condensed the Worldwide classes to directly compare Worldwide and CoNLL, for OntoNotes, we condensed the OntoNotes classes to match the new class set. See Appendix E for a complete explanation. Next, we retrain the Stanza model for OntoNotes and test on Worldwide, then train a model using the Worldwide dataset and test against OntoNotes, and finally train a combined model and test on both datasets.

The performance degradation observed for a

| spaCy model | OntoNotes (collapsed) | Worldwide | Africa | Asia | Indigenous | Latin America | Middle East |
|---|---|---|---|---|---|---|---|
| en_core_web_sm | 84.51 | 57.22 | 58.69 | 60.12 | 41.91 | 55.46 | 57.07 |
| en_core_web_trf | 91.72 | 76.25 | 77.78 | 79.19 | 67.58 | 74.48 | 73.21 |

Table 8: Entity F1 Scores for spaCy on OntoNotes and Worldwide

CoNLL model labeling the Worldwide dataset continues when using an OntoNotes model on the Worldwide dataset. Under the experimental setup described above, the best performing model trained on OntoNotes, using Electra-Large as the bottom layer, scored 92.55 F1 on entities. On the Worldwide dataset, though, it scored 77.00. Conversely, the Electra-Large model trained on Worldwide scored 88.97 on the Worldwide test set, but 77.86 on the OntoNotes. As seen with both Stanza and CoreNLP on CoNLL, though, a model trained on both performed well on both, scoring 87.79 on the Worldwide test set and 91.62 on OntoNotes.

A challenging entity type for the OntoNotes models to classify was Facility. The Electra-Large model trained on OntoNotes guessed 30% of the Facility tokens to be Organization, for example, and 36% of the time when it predicted Facility, the true label was Location. An example of this error is "Ras Dashen, the tallest mountain in Ethiopia", which the OntoNotes-trained Electra-Large model labels Facility, despite another mountain, Everest, being labeled Location in OntoNotes.

Again, these scores will differ from those in the literature due to some of the classes left out. Furthermore, some performance loss in the combined dataset is from issues with the word "the" or similarly mundane differences in labeling schemes. In CoNLL and our dataset, "the" is rarely labeled as part of an entity, whereas OntoNotes frequently labels "the" in entities such as "*the* White House".

### 7.4 SpaCy on OntoNotes

Another commonly used NLP framework is spaCy, which has both a transformer and a non-transformer model (Honnibal et al., 2020). As these models are trained on OntoNotes, we follow the same class reduction as for Stanza.

The spaCy model which does not use transformers, en_core_web_sm, shows a very large drop in performance; see Table 8. This is especially true on the Indigenous portion of the dataset, as we have observed in other test settings, except here the degra-

dation is more severe. The en_core_web_trf model shows a smaller drop, although it still performs comparatively worse on the Worldwide dataset than the Stanza models, which were retrained with the Worldwide training data. We conclude that the spaCy models are also affected by the regional variations in named entities. SpaCy does relatively better in Africa and is most affected in the Indigenous, Latin America, and Middle East contexts.

## 8 Conclusion

We introduce an NER dataset composed of newswire from outside Britain and America, designed to act as an "adversarial" evaluation set for English NER models solely trained on American or British contexts. We find limitations in Stanza's NER when trained solely on CoNLL or OntoNotes and tested in global contexts of English (10 percent drop in F1 test scores). The model becomes most confused when contextual clues, which often disambiguate named entities, are missing. However, context clues are not a panacea—we also find that unknown tokens from the Worldwide English contexts can cause confusion despite clear sentence context. These effects persist with the addition of pre-trained word embeddings from word2vec, RoBERTa, and ELECTRA, suggesting that large models are not a catch-all solution to the challenge of classifying regional contexts. With that said, we find that the American and British English-trained model performs better on global texts with the addition of BERT embeddings than without them. One takeaway from this study is that one way to achieve stronger NER performance across a diverse set of English contexts is to train on more diverse sets of English data. Indeed, we find that the combined model demonstrates strong performance on both datasets. Therefore, we propose that existing state-of-the-art models would do well to combine their data with similarly sourced global English corpora.

## Limitations

First, the loss in performance appears to be related to incomplete vocabulary, which means that even if models are trained on increasingly diverse and large sets of data, there will still always be more named entities from novel language contexts that are missing. Hence, we observe that the difficulties with English NER in contexts outside the US or the UK are rooted in the incompleteness of training data, which is an infinitely-scaling problem. However, we believe that given our findings, the inclusion of even a small dataset of diversely-sourced data is well worth the effort. Second, we admit the great amount of experimentation left to be completed. With larger datasets, we expect model performance on this dataset to improve dramatically compared to solely training on CoNLL03, but we have not attempted to test models trained on larger American or British English dominated datasets on the Worldwide dataset. We plan to conduct this research in coming months to understand if modern scaled models are fit to perform NER in global English language contexts. Lastly, we cannot speak to how this effect may manifest in other languages, but it would be interesting how the concept we investigate applies to languages with a vast diaspora or high worldwide uptake. French, Spanish, Dutch, or any other language considered a primary language in several countries may have similar issues; indeed, there are often separate research tools for Portuguese from Portugal or Brazil because of exactly the sort of effect studied in this paper.

## Ethics Statement

We believe that to confidently assert that NER is a task that is more-or-less solved in English, state-of-the-art models should be able to achieve strong performance in all English contexts, not just the most popular ones. After all, English is a language spoken widely across the world, not only in the lands in which it originated. Hence, we find that the ethical implications of our work are to bring stronger equity to the performance of English NER models. Additionally, since we find little related work in English NER, it is likely that NER in other languages has a lack of research with this idea as well. For example, regional dialects or contexts of languages such as Mandarin Chinese may use named entities in different ways than we traditionally believe, or use entirely different named entities. We hope that this work may inspire others to investigate this question in other languages as well to broaden the ethical impact of our work beyond English.

An important question when using labeling services is whether or not the annotation is ethically sourced. We note that our labeling workforce was composed of native English speakers from an overseas country as an alternative to student labelers. To the best of our knowledge, with consultation with Datasaur and MLTwist, the annotator team was paid a fair wage relative to the local market. Furthermore, we directly contacted Aya Data, the subcontractor in Ghana, and they responded that they enforce the following rules which they believe ensure the annotators are treated fairly:

- Pay annotation employees an average of >5x min wage in Ghana

- Employ all staff full time, with pensions paid. This means job security and no project work / zero hours contracts

- Don't accept any explicit content moderation projects, or anything that will expose employees to harmful content

- Regular training in non-annotation skills, and structured career paths

We believe this indicates the team was treated fairly and the dataset can be considered ethically sourced.

## Acknowledgements

Christopher Manning is a CIFAR Fellow. We would like to thank the annotation team at Datasaur, MLTwist, and Aya Data for their hard work. We would like to thank Percy Liang, Jeremy Cole, Shikhar Murty, and the Stanford NLP group for the helpful discussions we had of this paper. The anonymous reviewers each gave several valuable suggestions that strengthened the arguments made here. Finally, we would like to thank our friends and family for their support of our work.

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

## A Named Entity Classes

| Label | CoNLL | Description | Example |
|---|---|---|---|
| Person | PER | Names of humans, excluding deities (e.g. "God") | Ryan |
| Organization | ORG | Names of a group or collective body | Supreme Court |
| Location | LOC | A physical place | Paris |
| Facility | LOC | A place that is used for a specific utility | Brooklyn Bridge |
| NORP | MISC | A national, organizational, religious, or political identity | Chinese |
| Money | MISC | The name of a particular denomination of money | Yen |
| Product | MISC | The name of a marketable item or brand | Ford F-150 |
| Miscellaneous | MISC | Other named entities: events, works of art, etc | Mona Lisa |
| Date | O | Any time of day, month, or year | 2/19/2023 |
| O | O | Not a named entity | water bottle |

Table 9: Named entity classes, with definitions and examples

## B Dataset Statistics

| Region | Arts | Tokens | Date | Fac | Loc | Misc | Money | NORP | Org | Per | Prod |
|---|---|---|---|---|---|---|---|---|---|---|---|
| Africa | 246 | 153305 | 1445 | 349 | 3659 | 2268 | 541 | 1131 | 4692 | 4236 | 101 |
| Asia | 346 | 210295 | 2240 | 543 | 5604 | 2603 | 1166 | 1647 | 7245 | 6443 | 292 |
| Indigenous | 87 | 67065 | 489 | 150 | 1415 | 676 | 114 | 835 | 1410 | 1507 | 4 |
| Latam | 232 | 156162 | 1542 | 388 | 4773 | 1726 | 651 | 1065 | 4048 | 4045 | 70 |
| Middle East | 164 | 87984 | 517 | 134 | 2579 | 858 | 257 | 889 | 3006 | 3009 | 98 |

Table 10: Statistics by Region. Countries of Origin are further broken down at https://github.com/stanfordnlp/en-worldwide-newswire

| | Arts | Tokens | Date | Fac | Loc | Misc | Money | NORP | Org | Per | Prod |
|---|---|---|---|---|---|---|---|---|---|---|---|
| Train | 745 | 466130 | 4432 | 1112 | 12449 | 5637 | 2052 | 4035 | 14082 | 13416 | 496 |
| Dev | 100 | 64230 | 639 | 111 | 1705 | 863 | 207 | 403 | 2251 | 1894 | 31 |
| Test | 230 | 144451 | 1162 | 341 | 3876 | 1631 | 470 | 1129 | 4068 | 3930 | 38 |

Table 11: Statistics by Train/Dev/Test

## C Stanza Hyperparameters

| Hyperparameter | Value | Hyperparameter | Value |
|---|---|---|---|
| Optimizer | SGD | Word Embedding Size | 100 |
| Learning Rate | 0.1 | Character Embedding Size | 1024 |
| Learning Rate Decay | 0.5 | Gradient Clipping Max Norm | 5.0 |
| Early Termination | 0.0001 | Batch Size | 32 |
| LSTM Layers | 1 | Dropout | 0.5 |
| LSTM Hidden Layer Size | 256 | Word Dropout | 0.01 |

Table 12: Hyperparameters used for training the Stanza NER models

## D  Condensing Worldwide Labels

| Original label | Condensed label |
|---|---|
| Facility | Location |
| Work of Art | MISC |
| NORP | MISC |
| Currency | MISC |
| Product | MISC |
| Date | O |

Table 13: Condensed Worldwide labels into CoNLL 4-class set

An important remaining point of discrepancy between the datasets was our inclusion of dates, while CoNLL03 labels dates as O. To maintain consistency with CoNLL03, we process out all Date tags.

## E  Condensing OntoNotes Labels

| OntoNotes label | Condensed label |
|---|---|
| GPE | Location |
| Work of Art | MISC |
| Law | MISC |
| Event | MISC |
| Language | MISC |
| Cardinal | O |
| Ordinal | O |
| Percent | O |
| Quantity | O |
| Time | O |
| Date | O |

Table 14: Condensed OntoNotes labels into Worldwide class set

We note that labels such as Cardinal were ignored because they were not annotated in our Worldwide dataset. Additionally, due to differences in how Date is annotated in OntoNotes and our dataset, it is ignored as well. In OntoNotes, unnamed lengths of time are labeled, whereas Worldwide does not label such expressions.