# OpenReview forum: "Do “English” Named Entity Recognizers Work Well on Global Englishes?"
_EMNLP/2023/Conference — EMNLP 2023 Findings_

### Official Review · Reviewer_cg6A · 2023-08-04

**Soundness:** 4

**Excitement:**

4: Strong: This paper deepens the understanding of some phenomenon or lowers the barriers to an existing research direction.

**Missing References:**

n/a

**Paper Topic And Main Contributions:**

This paper asks whether Named Entity Recognition (NER) taggers trained on American/British English datasets for the task generalize to other dialects of English. To test this, it introduces a new NER dataset that contains texts from news sources from many different regions of the world where English is spoken. This dataset is then tested on many standard NER systems trained on one of two popular NER datasets (CONLL-03 and OntoNotes). The experiments show that models trained on these datasets perform much worse than models trained specifically for this setting, though the contextualized models generalize better than the non-contextualized ones. They also show that training the NER tagger with a combination of the original training data and the new data allows the models to generalize without losing performance on the original dataset.

The contributions of this paper are as follows:
- Introduces a new NER resource covering many dialects of English that previously did not have any data for this task.
- Demonstrates systems trained on existing NER datasets are not able to generalize to these new evaluation settings.
- Show that incorporating data from these dialects into the training process for these systems improves generalization to these dialects while also achieving similar performance on the original dataset's evaluation set.

**Questions For The Authors:**

[A] Can you elaborate on some of the missing details about the data annotation process? Specifically, how many annotators were there (per region)? Are the annotators native speakers of the dialect being annotated? Do you have any measures of inter-annotator agreement?

[B] Line 156: I think these data statistics are missing -- could you include a pointer to them in the text? It would be helpful to know the breakdown across train/dev/test for the new dataset, as well as within each region.

[C] Line 238-240: how were these hyperparameters chosen?

**Reasons To Accept:**

- The work points out an important flaw in the claim that English NLP is solved when this is not the case for non-standard dialects of English. It then documents the issue on a specific task (NER).
- The paper also points to a generally applicable method for improving models on non-standard dialects -- being cognizant of these data variations and including them in your data annotation and training process.
- They annotate and release a new NER dataset on lower-resource English language settings.
- The paper is clearly written and easy to read.

**Reasons To Reject:**

- A number of details about the data annotation process are not included in the paper (see Question A in the Questions to Authors)
- All of the analysis presented in this work is qualitative (Section 5). While these insights are interesting, it would be helpful to provide some quantitative analysis of the data and model performance as well. For example, the paper could include statistics about the types of errors discussed in the text across the different regions in the dataset; another interesting analysis might be the overlap of named entities between the original NER corpora and the new dataset and how that related to performance, as well as the distribution of new/unknown entities across regions of the data.

**Reproducibility:**

4: Could mostly reproduce the results, but there may be some variation because of sample variance or minor variations in their interpretation of the protocol or method.

**Reviewer Confidence:**

4: Quite sure. I tried to check the important points carefully. It's unlikely, though conceivable, that I missed something that should affect my ratings.

**Typos Grammar Style And Presentation Improvements:**

- It would be helpful to make some of the section titles more specific, particularly in the case of Sections 4 and 6, since they cover very similar topics.
- Lines 269-275: A combined model is discussed here, but it isn't included in Table 1. It would be helpful to add it to the table or include a pointer in the text to the model you are discussing.

---

> ### Author Rebuttal · Authors · 2023-08-29
>
> Thank you for your kind comments emphasizing the value of the new resource introduced and the easy-to-read paper.
>
> In the criticism section, you ask for more description of the dataset itself, as well as ask where the promised table on line 156 is.  Regrettably, it appears that it was omitted from the submitted draft of the paper.  We have recreated the statistics in our answer to Reviewer nGXt.
>
> As per the previous response, we can also clarify the data annotation process.  The question of whether or not the annotators are from the regions in question is an interesting one.  As it turned out, the annotators were all either in Ghana or the US.   If we had sourced the work for each article to annotators from the country in question, that would have improved the overall quality of the annotations, but this would be impractical considering the variety of regions and countries we used.  Nevertheless, we think we have a dataset with high quality annotations.
>
> Hyperparameters were chosen by hill climbing on the validation set of the CoNLL dataset, then reused for each training pass.  The purpose was not to find the absolute best parameters for each dataset, but to compare the scores with and without the regional data.  We can put a table of the hyperparameters in a final draft of the paper.
>
> The work creating the models should be very reproducible.  The conversion scripts to join our dataset with CoNLL, or OntoNotes with our dataset, will both be made public as part of the release of the dataset.  Furthermore, we used the default settings when retraining both CoreNLP and Stanza, so the models themselves can be recreated as needed.
>
> Thank you for the suggestion of including more quantitative analysis of the experiments.  We can certainly do that, such as by giving statistics on the types of errors per region.
>
> I would also say something about thanks for the suggestion of including more quantitative analysis of the experiments and say that we will try to do that, such as by giving the statistics on types of errors by region.
>
> The question of inter-annotator agreement is a good one.  We present the following discussion of the annotation process, along with Inter Annotator Agreement and dataset transparency.
>
> We labeled data using Datasaur’s labeling platform, from where we outsourced labeling management to MLTwist, an ML labeling firm. MLTwist used native English speakers, mostly in Ghana, who did not necessarily have dialect-specific knowledge for each language region we used. An overwhelming majority of the labelers were English speakers from Africa. As mentioned above, it would have been ideal to have representation from all of the regions we labeled, but that would not have been practical.
>
> We collected inter-annotator agreement scores from Datasaur, the data labeling platform used to create our dataset. We note that the results found do not measure true IAA between two independent annotators, but rather an index for correction from reviewers (see process below). Datasaur computes inter annotator agreement scores using Cohen’s Kappa, while taking into account the possibility of chance agreement.
>
> Overall, we employed 7 labelers. Our labeling system structure was composed of having one annotator per file, with a team of 2–3 reviewers for each file. The annotator was instructed to label all of the named entities in the articles, before the reviewers (at least one representative of MLTwist and at least one paper author) would confirm or adjust the initial annotator’s labels. When reviewing, the MLTwist and Stanford reviewer would communicate to achieve a consensus decision which is why the Datasaur platform computes agreement between the whole reviewer team and the annotator, rather than comparing per reviewer. Hence, the inter annotator agreement score we compute is between the reviewer team (MLTwist + Stanford) and a single annotator.
>
> The following table maps annotators (whose identities have been redacted for privacy) to their inter-annotator score. We computed an average inter-annotator score by averaging the score for each labeling batch. Finally, we also compute a joint inter-annotator agreement score (representing the average of all labelers) by averaging all inter-annotator scores.
>
> Average inter-annotator agreement scores, using Cohen's Kappa.  Approximately 50 articles per batch
>
>
> | Annotator ID | Cohen's Kappa | Batches annotated |
> | :----: | :----: | :----: |
> | 1 | 73.95 | 8 |
> | 2 | 83.52 | 3 |
> | 3 | 68.43 | 3 |
> | 4 | 79.53 | 5 |
> | 5 | 77.74 | 1 |
> | 6 | 77.87 | 1 |
> | 7 | 90.57 | 2 |
> | Joint | 77.47 | 23 |
>
> Unlike many reported NLP inter-annotator agreement scores which are simply percentage agreement, we measure using Cohen’s Kappa. Furthermore, we find that the agreement scores between reviewers and the annotators were quite good. While scales for Cohen’s Kappa are arbitrary and heavily context-dependent, we feel that our results reflect a thorough labeling process.
>
> Thank you for your consideration and helpful suggestions to improve our paper for the final revision. We will be sure to include additional explanations surrounding the annotation process in the final revision.

---

### Official Review · Reviewer_nGXt · 2023-08-04

**Soundness:** 4

**Excitement:**

3: Ambivalent: It has merits (e.g., it reports state-of-the-art results, the idea is nice), but there are key weaknesses (e.g., it describes incremental work), and it can significantly benefit from another round of revision. However, I won't object to accepting it if my co-reviewers champion it.

**Paper Topic And Main Contributions:**

This paper introduces a new dataset for English NER that focuses on English from other countries outside of the US and UK. The main claim is that the current NER datasets and models are UK- and US-centric, and do not perform equally good on English languages from other parts of the world. The authors create a training and test
dataset built from newswire from outside the US and Europe containing 1075 articles comprising 674,000 tokens. These articles are from countries and regions across the world where English is used. The authors also divide the texts into regional buckets: Asia, Africa, Latin America, the Middle East, and Indigenous Commonwealth (indigenous Oceania and Canada). While the dataset is fairly described, very little statistical information and details about the sources are provided.
The dataset is annotated for NER using crowdsourcing. However, no inter annotator agreement is provided.

Once the dataset created, the authors move on to test a selection of NER toolkits and transformer models on CoNLL 2003 and OntoNotes5.0 with the new dataset. The tested toolkits are Stanza, CoreNLP, Flair, and Spacy. They show that the models trained on the CoNLL and OntoNotes datasets drop performance on the newly created dataset. They also observe the opposite, that models trained on the new dataset perform worse on CoNLL and OntoNotes. However, combining the datasets with the diverse English data gives better performance both for US and UK English and on English languages spoken elsewhere.

While the paper is well written and have sound arguments, some details are lacking about the annotation process and the dataset created. Since this is the main contribution, I see this as a serious flaw. Moreover, in the ethical section the authors do not discuss the ethical aspect of how they have collected their dataset (which is also lacking in the discussion in the paper).

**Reasons To Accept:**

- Important work on making available tools work on other English languages outside the US and the UK.
- Nice error analysis and discussion of the results.

**Reasons To Reject:**

- Details about the annotation process are lacking.
- Details about the dataset are lacking.
- No ethical discussions are mentioned about how the dataset has been collected.

**Reproducibility:**

3: Could reproduce the results with some difficulty. The settings of parameters are underspecified or subjectively determined; the training/evaluation data are not widely available.

**Reviewer Confidence:**

3: Pretty sure, but there's a chance I missed something. Although I have a good feel for this area in general, I did not carefully check the paper's details, e.g., the math, experimental design, or novelty.

---

> ### Author Rebuttal · Authors · 2023-08-28
>
> Thank you for your helpful comment and for finding the work well-written with nice error analysis and utility for studying worldwide Englishes.
>
> The annotation process was as follows.  First, the authors collected the articles from a variety of international websites and extracted the text.  We then sent this off to Datasaur and their partner, MLTwist, for annotation.  MLTwist's process was to have one annotator review each document, then have a final authority review the annotations and correct any discrepancies found.  The authors then reviewed the annotations ourselves, especially in cases where a model trained on the training set then reported errors on its own training set, and corrected any incorrect labels we found.  We can clarify this for the final version of the paper.
>
> MLTwist's annotation team was in Ghana, which may raise the ethical question of whether or not the work was exploitative or if the annotators were paid a fair wage.  To the best of our knowledge, the annotator team was paid a fair wage relative to the local market.  This issue was briefly discussed at the end of the Ethics Statement in the submitted paper.  To further answer this question, we followed up with the annotation team, and we heard directly from the subcontractor in Ghana that they enforce the following rules which they believe ensure the annotators are treated fairly:
>
> 1) Pay annotation employees an average of >5x min wage in Ghana
>
> 2) Employ all staff full time, with pensions paid. This means job security and no project work / zero hours contracts
>
> 3) We don't accept any explicit content moderation projects, or anything that will expose our employees to harmful content
>
> 4) Regular training in non-annotation skills, and structured career paths
>
> The work creating the models should be very reproducible.  The conversion scripts to join our dataset with CoNLL, or OntoNotes with our dataset, will both be made public as part of the release of the dataset.  Furthermore, we used the default settings when retraining both CoreNLP and Stanza, so the models themselves can be recreated as needed.
>
> Thank you for the suggestion to give more details on statistics of the dataset. We are happy to do this and include some below.
>
> Statistics for each region of the dataset
>
> | Region | Tokens | Date | Facility | Location | Misc | Money | NORP | Org | Person | Product |
> | :--- | :---: | :---: | :---: | :---: | :---: | :---: | :---: | :---: | :---: | :---: |
> | Africa | 153305 | 1445 |349 |3659 |2268 |541 |1131 |4692 |4236 |101 |
> | Asia | 210295 | 2240 | 543 | 5604 | 2603 | 1166 |1647 |7245 |6443 |292 |
> | Indigenous | 67065  | 489 |150 |1415 |676 |114 |835 |1410 |1507 |4 |
> | Latam | 156162 | 1542 | 388 | 4773 | 1726 |651 |1065 |4048 |4045 | 70 |
> | Middle East | 87984 | 517 | 134 | 2579 | 858 | 257 |889 | 3006 | 3009 | 98|
>
> Divided by train/dev/test
>
> |  | Tokens | Date | Facility | Location | Misc | Money | NORP | Org | Person | Product |
> | :--- | :---: | :---: | :---: | :---: | :---: | :---: | :---: | :---: | :---: | :---: |
> | Train | 466130 | 4432 | 1112 | 12449 | 5637 | 2052 | 4035 | 14082 | 13416 | 496 |
> | Dev | 64230 | 639 | 111 | 1705 | 863 | 207 | 403 | 2251 | 1894 | 31 |
> | Test | 144451 | 1162 | 341 | 3876 | 1631 | 470 | 1129 | 4068 | 3930 | 38 |
>
>
> Thank you again for the valuable feedback regarding this work.

---

### Official Review · Reviewer_Mety · 2023-08-09

**Soundness:** 3

**Excitement:**

3: Ambivalent: It has merits (e.g., it reports state-of-the-art results, the idea is nice), but there are key weaknesses (e.g., it describes incremental work), and it can significantly benefit from another round of revision. However, I won't object to accepting it if my co-reviewers champion it.

**Paper Topic And Main Contributions:**

The authors have presented the Worldwide English NER dataset, a newly introduced corpus, to assess NER model performance in the context of "low-resource" English variations across different global regions. The datasets consist of five main categories, such as Africa, Asia, Indigenous, Latin America, and Middle East. The experiment results demonstrated that current NER models cannot handle English variations, although using a transformer model.

**Questions For The Authors:**

Did the authors have a chance to experiment with multilingual models? (XLM-R or mBERT). Given that these models are trained on diverse multilingual datasets, they inherently possess a higher potential to mitigate challenges associated with English variants.

**Reasons To Accept:**

1. The paper is well written, providing a clear motivation and view of the comparative landscape.
2. The outcomes presented within the paper are indeed promising
3. The authors provide a comprehensive ablation analysis showing the impact of English variants.
4. Additionally, the authors have introduced novel datasets for experimentation, thereby enriching the research's empirical foundation.

**Reasons To Reject:**

1. There appear to be potential issues within the experimental settings:

1a. Unseen Entity Problems: The authors highlight a performance drop when training a NER model on CoNLL and subsequently testing on the Worldwide dataset. This discrepancy could potentially be attributed to the presence of unseen entities, given the substantial temporal gap between the formulation of CoNLL (1996-1997) and the Worldwide dataset (2020s). There might be a lot of unseen entities in the test set, and the authors might further delve into this phenomenon, exploring the possibility of numerous unseen entities in the train/test set.

1b. Domain/Annotation Shift Problems: Building upon the observation in 1a, it's plausible that the performance drop is from a domain mismatch rather than the variations in English. Table 5 underscores this with OntoNotes, a dataset comprising diverse text genres (news, conversational telephone speech, weblogs, usenet newsgroups, broadcast, and talk shows). A notable contrast exists between the varied genres in OntoNotes and the Worldwide dataset, suggesting that the performance drop might be due to domain shift rather than English variants.

1c. Duplicate Problems (Data Leak): Considering the Worldwide datasets' compilation from global news, there's a potential for duplicate news items from distinct regions. For example, news about the US election might appear in Asia, Latin America, and Africa datasets. We might put the US news from Asia into the training set but split the US news from Latin America and Africa into the test set. Consequently, training a NER model on the Worldwide datasets could inadvertently result in improved performance within their dataset.

2. The ablation studies shed some light on the significance of considering diverse English variants. However, the paper falls short in offering comprehensive guidance on effectively managing these variations, apart from the conventional combined strategy. It would benefit the authors to delve into a more detailed analysis, providing insights into potential strategies for addressing this challenge.

**Reproducibility:**

4: Could mostly reproduce the results, but there may be some variation because of sample variance or minor variations in their interpretation of the protocol or method.

**Reviewer Confidence:**

3: Pretty sure, but there's a chance I missed something. Although I have a good feel for this area in general, I did not carefully check the paper's details, e.g., the math, experimental design, or novelty.

---

> ### Author Rebuttal · Authors · 2023-08-28
>
> Thank you for your thoughtful review; we were encouraged that you found the paper well written, providing a useful new resource with comprehensive evaluations.
>
> The criticism of the experimental settings is a very good point.  To summarize our interpretation of the argument, the variation in regional English may not be the only cause of the discrepancy in the model scores when compared on the WorldWide dataset.  Other explanations would include different domains, annotation differences, or temporal drift.
>
> The domain of the new dataset is newswire, as was the domain of both CoNLL and the majority of OntoNotes.  Because the original models for CoreNLP, Stanza, spAcy, and Flair were trained on one or the other dataset, those models should perform best on newswire data.  Therefore, the domain of the text should not explain the differences in performance we report.
>
> The concern over temporal drift is entirely valid. Thank you for raising it; we should have discussed it.  We can refer to another recent work on modern NER taggers, "Do CoNLL-2003 Named Entity Taggers Still Work Well in 2023?" by Shuheng Liu and Alan Ritter.  In this work, they present a case that transformer models such as Roberta do not show much temporal drift.  On the other hand, in our work, we find that a Roberta model loses performance when trained on CoNLL and tested on WorldWide, whereas training on the combined data maintains a high level of performance.
>
> Liu and Ritter tests for temporal drift by creating a smaller gold labeled dataset of newswire from 2020 and using it as a test set for a variety of models.  As an additional experiment, we instead added that to the CoNLL training data to see if a model trained on CoNLL + 2020 newswire would perform better on the WorldWide dataset.  The results are
>
> Retraining Stanza with a Roberta transformer
>
> | Training | CoNLL result | WorldWide result |
> | :--- | :---: | :---: |
> | CoNLL | 92.54 | 83.29 |
> | CoNLL + WorldWide | 91.84 | 89.40 |
> | CoNLL + 2020 | 92.52 | 84.57 |
> | CoNLL + 2020 + WorldWide | 91.95 | 89.75 |
>
> There is some gain from adding more recent training data, but not nearly as much as adding the entire WorldWide training set.  We believe this demonstrates that temporal drift alone does not explain much of the performance degradation we observed in our paper.  Iif this paper were to be accepted, we would be certain to add this result and the corresponding citation in order to address this question.
>
> Thank you for the good suggestion on the ways we could extend the ablation study.  Since we have cross-sections of the training and test data by region, we could, for example, train on CoNLL plus a subset of the regions to see whether adding some of the regions improves scores on a final held out region's test set.  If it would address this concern, we can include these results as well in the final version of the paper.
>
>
> Retraining Stanza with a Roberta transformer
>
> | Training portions | Asia WorldWide result | Entity coverage |
> | :---  |  :---: | :---: |
> | CoNLL | 85.19 | 49.41 |
> | CoNLL + Latam | 87.75 | 56.75 |
> | CoNLL + Latam + Africa | 88.18 | 59.33 |
> | CoNLL + Latam + Africa + Middle East | 88.06 | 61.83 |
> | CoNLL + Latam + Africa + Middle East + Indigenous | 88.51 | 62.18 |
> | CoNLL + only Asia | 90.36 | 73.85 |
> | CoNLL + entire WorldWide | 92.18 | 75.90 |
>
> Again we observe that adding some recent training data, starting with Latin America and continuing to add more, somewhat improves the results on the Asia test section of the dataset, going from 85.19 to 88.51, similar to adding the 2020 newswire from Liu and Ritter.  However, the largest gain is training on CoNLL plus the specific region in question, Asia, again indicating that simply having additional recent data is not as important as having localized data.
>
> The question regarding possible data leakage is a good one.  In general, there may be some test set leakage within a region, as it is possible multiple news articles covered the same incident and that went unnoticed by the authors, as multiple people collected the articles to be labeled.  However, the rate of this should be very low and it is highly unlikely there is leakage from region to region. The reason for this is that we sought out local news rather than global news from each publication, to try to ensure that we were getting locally written text, rather than repackaged material from international newswires.
>
> The specific question of whether xlm-roberta-large or mbert has less degradation is also an interesting topic to consider.  In initial experiments, we found that English Roberta and Electra-Large had better performance than the multilingual models, so we performed the final work using exclusively those models.  Running a subset of the experiments now, we find that
>
>
> Stanza retraining
>
> | Transformer | Training Portion | CoNLL F1 | WorldWide F1 |
> | :--- | :---: | :---: | :---: |
> | xml-roberta-large | CoNLL | 92.54 | 82.53 |
> | mbert | CoNLL | 92.09 | 81.44 |
> | xml-roberta-large | WorldWide | 75.46 | 90.25 |
> | mbert | WorldWide | 75.32 | 89.20|
>
> These numbers are all very similar to the numbers for Electra and Roberta.  A potential reason we did not see improved performance despite having a more international transformer model could be that although there is more international data in those transformers, it was not present in English contexts, so the prediction head on top of the transformer does not recognize the contextual embeddings of those words in English contexts.  Another possible explanation is that Roberta-Large already includes CC-News in its training data, which contains a wide variety of newswire, meaning that adding more non-English sources does not actually broaden the international applicability of the other transformers as much as expected. We will include this possible approach and results from it in a revised version of the paper.
>
> Your suggestions and questions for the authors have led to several experiments which we believe strengthen our case that it is regional variations causing low performance for models trained only on CoNLL Thank you again for your consideration.

---

### Meta-Review · Area_Chair_pses · 2023-09-15

**Recommendation:** 4

**Metareview:**

This paper introduces a new dataset for English NER that focuses on English varieties different from US and UK English. The authors then test various existing NER tools and models to evaluate their robustness to non-standard varieties.

The reviewers found that this paper addresses an important gap in current NLP that focuses on the varieties of English with the most (native) speakers. The error analysis was also found to be instructive. In contrast, the reviewers mentioned that some confounding factors (temporal drift etc.) were insufficiently taken into account, and that the paper lacked details about the annotation process, the statistics of the resulting dataset, and potential ethical issues in connection with the annotation. The authors addressed all these points in their rebuttals and promised to add them to the paper. As a result, all three reviewers were able to raise their scores.

---

### Decision · Program_Chairs · 2023-10-07

**Decision:**

Accept-Findings

**Comment:**

This paper introduces a new dataset for English NER that focuses on English varieties different from US and UK English. The authors then test various existing NER tools and models to evaluate their robustness to non-standard varieties.

The reviewers found that this paper addresses an important gap in current NLP that focuses on the varieties of English with the most (native) speakers. The error analysis was also found to be instructive. In contrast, the reviewers mentioned that some confounding factors (temporal drift etc.) were insufficiently taken into account, and that the paper lacked details about the annotation process, the statistics of the resulting dataset, and potential ethical issues in connection with the annotation. The authors addressed all these points in their rebuttals and promised to add them to the paper. As a result, all three reviewers were able to raise their scores.